# Field-free spin–orbit torque switching in ferromagnetic trilayers at sub-ns timescales

Qu Yang[1], Donghyeon Han[2], Shishun Zhao[1], Jaimin Kang [2], Fei Wang[1], Sung-Chul Lee[3], Jiayu Lei[1], Kyung-Jin Lee [4], Byong-Guk Park [2] & Hyunsoo Yang [1] ✉

Current-induced spin torques enable the electrical control of the magnetization with low energy consumption. Conventional magnetic random access memory (MRAM) devices rely on spin-transfer torque (STT), this however limits MRAM applications because of the nanoseconds incubation delay and associated endurance issues. A potential alternative to STT is spin-orbit torque (SOT). However, for practical, high-speed SOT devices, it must satisfy three conditions simultaneously, i.e., field-free switching at short current pulses, short incubation delay, and low switching current. Here, we demonstrate field-free SOT switching at sub-ns timescales in a CoFeB/Ti/CoFeB ferromagnetic trilayer, which satisfies all three conditions. In this trilayer, the bottom magnetic layer or its interface generates spin currents with polarizations in both in-plane and out-of-plane components. The in-plane component reduces the incubation time, while the out-of-plane component realizes field-free switching at a low current. Our results offer a field-free SOT solution for energy-efficient scalable MRAM applications.

Current-induced magnetization switching in nanomagnets has recently gained increasing attention to meet the demand for compact, fast, and energy-efficient magnetic storage cells and devices[1–4]. Spin transfer torques (STT)[1,2,5] are presently utilized for magnetic random access memory (MRAM) applications with the magnetization of the nanomagnet controlled by an electric current that passes through an oxide layer in magnetic tunnel junctions (MTJs). STT-MRAM has become a competitive technology as a replacement of embedded flash, and is promising for automotive and in-memory applications[6–8]. However, STT encounters limitations in ultrafast magnetization switching, which is of great importance for the development of high-speed data storage and processing devices. Time-resolved studies have shown that the switching speed of STT is limited to a timescale of 1 to 100 ns because of the non-negligible incubation delay induced by the collinear magnetization alignment of the reference and free layer[9–11]. Enforced fast switching of STT may be possible with a large current density ($J_c$) in the

writing process, however, this causes endurance issues, as the large $J_c$ degrades the oxide barrier of the MTJs. This renders STT-MRAM unsuitable for ultrafast applications such as cache memories. Moreover, the single path for both reading and writing makes it challenging to attain reliable reading operations[6].

To overcome the above limitations, spin-orbit torque (SOT) switching has been proposed as an alternative to the STT scheme[3,4,12–14]. In conventional SOT devices with perpendicular magnetic anisotropy (PMA) of the switchable free layer, the current-induced damping-like torque is orthogonal to the magnetization of the free layer. It thus minimizes the incubation delay during the switching and enables sub-nanosecond (sub-ns) switching with an assist magnetic field[15]. Moreover, the SOT scheme decouples the write and read current paths significantly enhancing reading reliability and device endurance as compared to the STT scheme[6,16]. However, the development of SOT-MRAM for ultrafast applications still remains challenging as an

[1]Department of Electrical and Computer Engineering, National University of Singapore, Singapore 117576, Singapore. [2]Department of Materials Science and Engineering, Korea Advanced Institute of Science and Technology (KAIST), Daejeon 34141, Korea. [3]Next Generation Process Development Team, Semiconductor R&D Center, Samsung Electronics Co. Ltd., Hwaseong, Gyeonggi 18448, Korea. [4]Department of Physics, Korea Advanced Institute of Science and Technology (KAIST), Daejeon 34141, Korea. ✉e-mail: eleyang@nus.edu.sg

in-plane (IP) magnetic field is typically required to achieve deterministic switching[4], which is an obstacle for practical SOT-MRAM applications. To tackle this issue, various approaches have been proposed to achieve field-free SOT switching, such as utilizing antiferromagnet[17,18], interlayer exchange coupling[19], tilted magnetic anisotropy[20], geometrical asymmetry[21,22], and out-of-plane spin accumulation[23,24]. Despite some progress, a clear field-free solution for ultrafast SOT operations remains elusive for real applications.

In this work, we demonstrate a field-free ultrafast SOT switching with CoFeB/Ti/CoFeB/MgO trilayers in which the top PMA CoFeB layer can be manipulated by spin currents generated from the bottom magnetic layer[25] or its interfaces[24,26]. An important difference of this trilayer from other field-free switching schemes is that spin polarizations carried by spin currents have both in-plane and out-of-plane components. It is expected that the in-plane component will help reduce the incubation delay analogous to the conventional SOT, while the out-of-plane component enables field-free switching at a low current[27]. We show that this expectation is indeed realized in magnetic trilayer structures. Sub-ns field-free switching of the top CoFeB magnetization is obtained with a $J_c$ 3 to 4 times lower than that reported for field-assisted pulse switching at sub-ns timescales[15,28,29]. The incubation time of the switching process is estimated to be 0.0144 ~ 0.226 ns when pulse duration $\tau_p$ ranges from 0.1 to 10 ns. This is much smaller than the STT incubation time which, despite attempts to improve performance, can be up to several tens of nanoseconds[30,31]. One-dimensional micromagnetic simulations indicate that the out-of-plane (spin-z) spin current in the ferromagnetic (FM) trilayer structure is essential in reducing the current density of the short pulse-induced SOT switching. The low writing current density and the intrinsic field-free characteristic make this ultrafast SOT switching demonstration promising for energy-efficient SOT applications.

## Results

To explore the field-free SOT-driven ultrafast magnetization switching, we prepare samples of Si/SiO$_2$ substrate/Ta (2 nm)/CoFeB (4 nm)/Ti ($t_{Ti}$ nm)/CoFeB (1 nm)/MgO (3.2 nm)/Ta (2 nm), where Ti thickness ($t_{Ti}$) varies from 1 to 4 nm. To induce IP magnetic anisotropy in the bottom CoFeB (4 nm) layer, a magnetic field of 15 mT is applied along the $x$ direction during the deposition. The Hall-bar-structured device with a round-shaped FM island of the fabricated trilayer sample is shown in Fig. 1a. The pillar is etched to the Ti layer only, and the bottom FM layer is fully covered across the Hall bar channel. The current flows in the $x$ direction with a bias tee separating the pulse of nanoseconds and the direct current ($I_{dc}$) (Fig. 1a). The perpendicular magnetization in the top CoFeB layer is probed via the anomalous Hall resistance ($R_{AHE}$) using a low $I_{dc}$. In the inset of Fig. 1b, the heterostructure stack with $t_{Ti}$ = 3 nm is shown. The top CoFeB layer (1 nm) is perpendicularly magnetized and serves as a spin current analyzer while the bottom IP-magnetized CoFeB layer or its interface is capable of generating the spin current to switch PMA

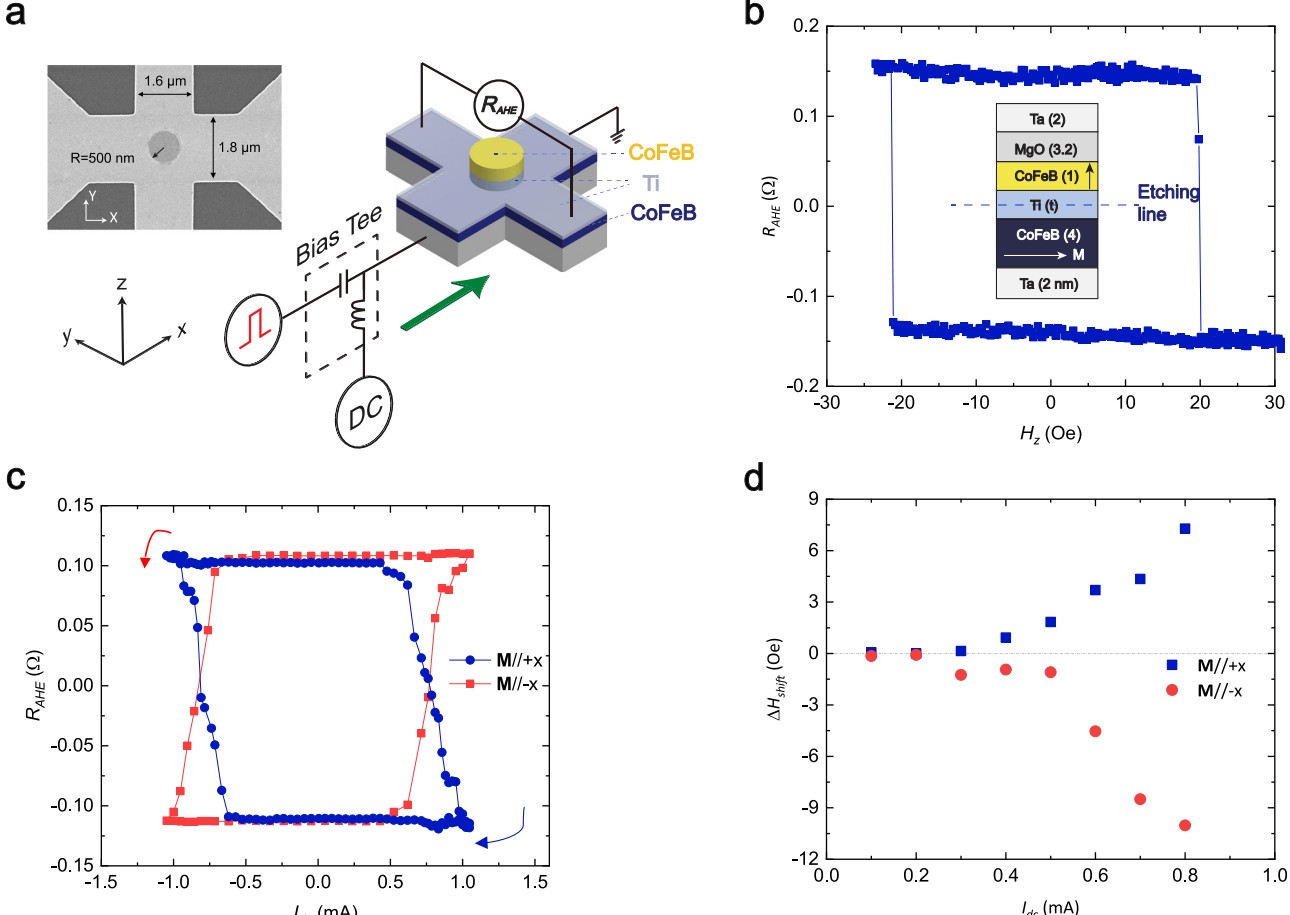

**Fig. 1 | SOT measurement of FM trilayers. a** Device image examined by scanning electron microscopy and the schematic illustration of the measurement setup is shown on the right side. **b** Magnetic hysteresis loop of the Hall bar device ($R$ = 500 nm) with the out-of-plane magnetic field ($H_z$). **M** indicates the magnetization direction of the bottom CoFeB layer. **c** Current induced field-free deterministic SOT switching in a $R$ = 500 nm pillar device when **M** is saturated along the +$x$ and −$x$ direction. **d** Current dependence of the anomalous Hall loop shift ($\Delta H_{shift}$) with **M**//+$x$ and **M**//−$x$ in a $R$ = 500 nm pillar device. The measurements shown in this figure are conducted on the sample with $t_{Ti}$ = 3 nm.

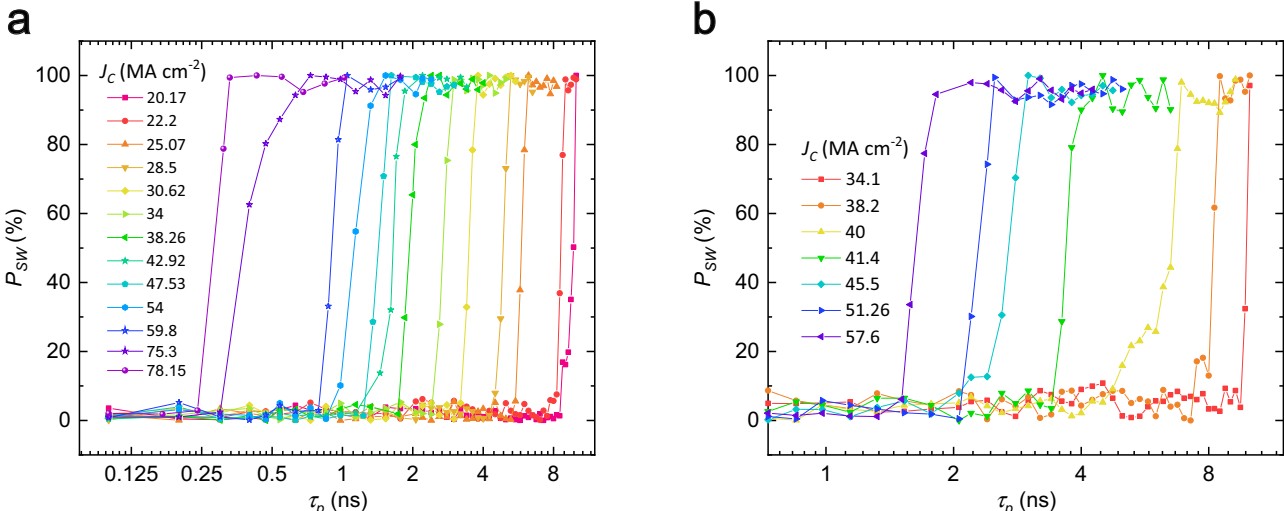

**Fig. 2 | Switching probability ($P_{sw}$) of field-free short pulse SOT switching in CoFeB/Ti (3 nm)/CoFeB trilayer with $R$ = 500 nm.** A single positive pulse (**a**) or negative pulse (**b**) is applied as a function of pulse duration ($\tau_p$) at different pulse current densities. **M** is saturated along the $-x$ direction.

**Table 1 | Comparison of short pulse SOT switching of PMA devices**

| Device Structure (nm) | Pulse width (ns) | Anisotropy field $B_{an}$ (mT) | $J_c$ (MA cm$^{-2}$) | Magnetic field (mT) | Write energy per area (mJ cm$^{-2}$) | Reference |
|---|---|---|---|---|---|---|
| Pt(3)/Co(0.6) | 0.3 | 1000 | 387.5 | 91 | 11.2 | 15 |
| W(3.7)/CoFeB(0.9) | 0.27 | 270 | 312 | 23 | 8 | 28 |
| W/CoFeB(0.9) | 0.6 | -- | 276 | 40* | 19.1 | 37 |
| Ta(10)/CoFeB(1) | 0.4 | -- | 340 | 100 | 2971 | 29 |
| CoFeB(4)/Ti(3)/CoFeB(1) | 0.14 | 400 | 73.2 | 0 | 1.51 | This work |

*A 50 nm thick Co magnet was added on top of the device to produce an in-plane field of 40 mT.

without any assist field[24]. The stray field generated by the bottom CoFeB layer on the top FM layer was estimated to be -0.1 mT. This value is significantly smaller than the effective in-plane magnetic field of 2.7 mT generated by the $z$-spin current in the magnetic trilayer structures[32].

The magnetic hysteresis loop of the trilayer device with a field sweeping along the out-of-plane direction signifies the presence of good PMA in the top CoFeB layer (Fig. 1b). We first characterize the DC switching properties. Figure 1c shows the $I_{dc}$ induced field-free SOT switching with $J_c$ = 12 MA cm$^{-2}$. When the magnetization (**M**) of the in-plane bottom layer points along the $+x$ and $-x$ direction, the switching polarity is clockwise and counterclockwise, respectively. The saturation direction of **M** establishes the reference state for the magnetization configuration and does not affect the switching parameters such as $J_c$ and $\tau_p$. The incoming charge current in the bottom FM/non-magnet (NM) bilayer induces the spin polarization in the **y** direction as well as in the **M×y** direction[24]. For an IP-magnetized ferromagnet (**M**//±**x**), therefore, the **M×y** component is nothing but the spin $z$ component, which lead to field-free SOT switching of top PMA layer. This field-free switching behavior has been confirmed in different samples with $t_{Ti}$ = 1 ~ 4 nm (Supplementary Note 1), among which the sample with $t_{Ti}$ = 3 nm shows the most stable switching performance and the lowest $J_c$. To further confirm the existence of the spin $z$ component in the CoFeB/Ti/CoFeB trilayer structure, we characterize the out-of-plane SOT effective field by measuring the anomalous Hall loop shift ($\Delta H_{shift}$) under different $I_{dc}$ (Supplementary Note 2). As shown in Fig. 1d, the loop shift of the $t_{Ti}$ = 3 nm sample can be observed and $\Delta H_{shift}$ increases as $I_{dc}$ increases. The direction of the loop shift reverses when changing the direction of **M**, which is consistent with the previous work[24]. Based on the AHE loops under different $I_{dc}$, the existence of out-of-plane spin is confirmed experimentally.

In order to further study the ultrafast dynamics of the field-free SOT switching, we apply positive and negative short pulses and perform switching probability measurements on a trilayer pillar device with $t_{Ti}$ = 3 nm. The calibration configuration of equivalent $J_c$ with a short pulse across the pillar has been described in Supplementary Note 3. As shown in Fig. 2, the switching probability ($P_{sw}$) of the anomalous Hall signal is investigated with the pulse width ($\tau_p$) ranging from 0.1 to 10 ns. Notably, the minimum pulse width required to achieve $P_{sw}$ = 0.9 for the field-free SOT switching can be as short as $\tau_p$ = 0.3 ns for $J_c$ = 78.15 MA cm$^{-2}$ at $R$ = 500 nm (shown in Fig. 2). With a reduced radius of $R$ = 300 nm, $\tau_p$ can be further decreased to just 0.14 ns (Table 1, Supplementary Fig. 6). In contrast, SOT switching of PMA in the CoFeB/MgO device of equivalent lateral dimension requires $\tau_p$ to be ~1 ns, when a similar current density is applied even in the presence of an assisted field of 119.1 mT[33]. Although SOT switching with a sub-ns pulse can also be achieved in some PMA structures, a very high current density of 300 ~ 400 MA cm$^{-2}$ in addition to assisted fields was required in previous studies for reliable deterministic switching[15,28,29]. We have summarized the results on SOT short pulse switching of PMA with previously reports in Table 1. Devices made from films with identical layer thicknesses maintain their reproducibility, as evidenced by the DC switching (Supplementary Note 4) and short pulse switching (Supplementary Note 5) with various dimensions ($R$ = 75 ~ 550 nm, Supplementary Note 6).

In comparison, the sub-ns pulse switching in our CoFeB/Ti/CoFeB trilayer requires a current pulse of much lower density than the field-assisted sub-ns switching. Consequently, the writing energy consumption per unit area of our trilayer FM device is low (1.51 mJ cm$^{-2}$ shown in Table 1), and the intrinsic field-free characteristic makes it promising for energy-efficient SOT-based applications. Field-free short pulse switching can be achieved with various $t_{Ti}$ as shown in Supplementary Fig. 8. DC switching typically exhibits a relatively low current density (e.g. $J_c$ = 12 MA cm$^{-2}$ in our $t_{Ti}$ = 3 nm sample) while higher

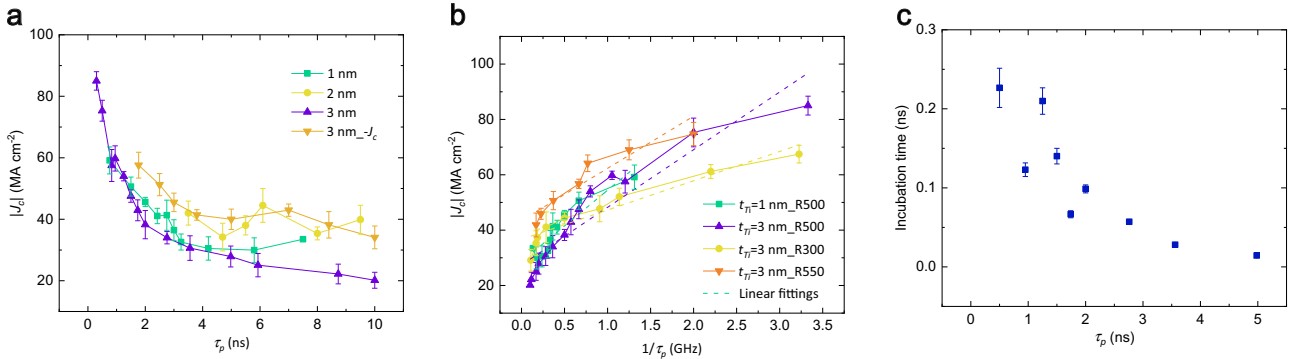

**Fig. 3 | Critical switching current density ($J_c$) and incubation time estimation for field-free SOT short pulse switching. a** $|J_c|$ for switching probability $P_{sw} = 0.9$ as a function of pulse duration ($\tau_p$) with $R = 500$ nm. **b** $|J_c|$ for $P_{sw} = 0.9$ as a function of $1/$ $\tau_p$ at different $t_{Ti}$ and dimensions (R = 300, 500, and 550 nm). **c** Incubation time of the positive pulse induced SOT switching as a function of $\tau_p$, with $t_{Ti} = 3$ nm with $R = 500$ nm (corresponding to Fig. 2a). Error bars represent the standard deviation.

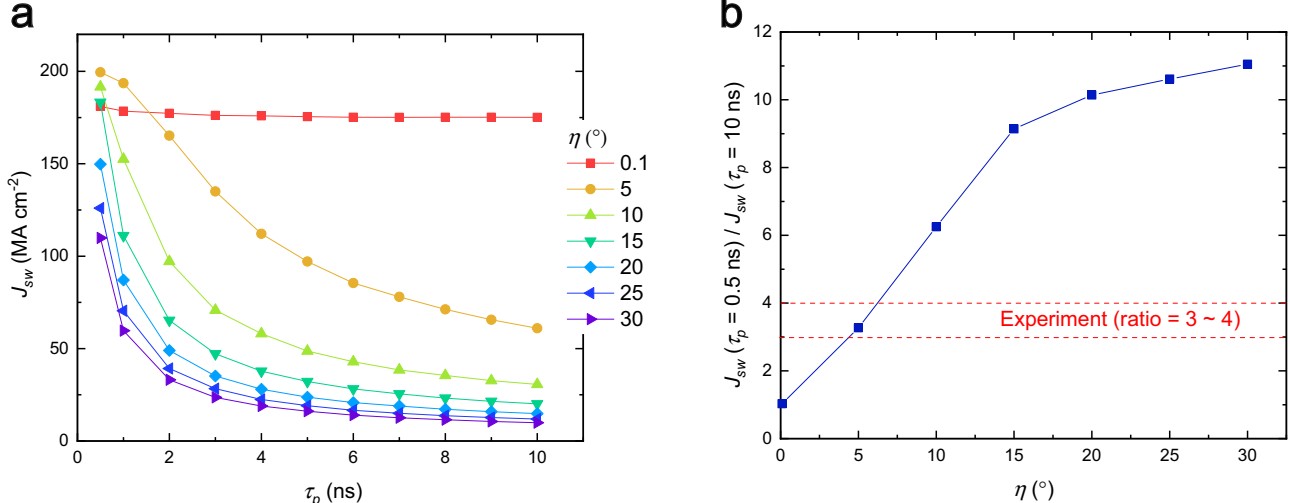

**Fig. 4 | Micromagnetic simulations of short pulse induced SOT switching. a** Switching current density ($J_{sw}$) as a function of pulse width ($\tau_p$) with different spin polarization angles η = 0.1 - 20°. **b** The ratio of $J_{sw}$ with $\tau_p = 0.5$ ns to $J_{sw}$ with $\tau_p = 10$ ns as a function of η.

current densities are required as the pulse width gradually decreases to the sub-ns region (shown in Supplementary Fig. 9).

The energy dissipation during the switching process is reflected by $J_c$ and $\tau_p$. Figure 3a shows the absolute value of $J_c$ as a function of $\tau_p$, which is defined at $P_{sw} = 0.9$. The pulse switching behavior has been reported to show two distinct regimes[15,34], a long-duration thermal-assisted regime and a short-duration intrinsic regime ($\tau_p < 10$ ns) in which $J_c$ increases dramatically with reduced $\tau_p$. We plot $1/\tau_p$ versus $|J_c|$ as shown in Fig. 3b for $\tau_p < 10$ ns, and fit to the model $I_c = I_{c0} + q/\tau_p$[15]. The parameter $I_{c0}$ refers to the intrinsic critical switching current needed to overcome the magnetization damping, and $q$ is an effective charge parameter describing the efficiency between charge and spin-angular momentum transport. The linear proportional dependence shown in Fig. 3b is a signature of the spin torque switching process[15]. In our case, the critical switching current density $J_{c0}$ is determined by dividing $I_{c0}$ with the cross-sectional area of devices. From the fitting shown in Fig. 3b, we obtain $J_{c0} = 27.41$ MA cm$^{-2}$ and $q = 4.95 \times 10^{-13}$ C for $t_{Ti} = 3$ nm, while $J_{c0} = 28.63$ MA cm$^{-2}$ and $q = 5.44 \times 10^{-13}$ C for $t_{Ti} = 1$ nm. Analytic models for finite temperature spin-torque dynamic have primarily focused on uniaxial single domain nanomagnets[15,34]. In our specific case, the device is more likely exhibiting multi-domain wall motion rather than reversing as a single magnetic domain, especially given its large lateral dimension ($R = 300 - 550$ nm). This characteristic may contribute to the observed deviations in the

linearity of the experimental results. As the dimensions decrease down to $R = 75$ nm, the linearity improves, as shown in Supplementary Note 7.

The switching probability shown in Fig. 2a can be theoretically fitted by the exponential function as described in Supplementary Note 10, through which the incubation time of the switching process is estimated to be 0.0144 - 0.226 ns as shown in Fig. 3c. It is important to note that the main portion of the switching time in our measurements is not the incubation time but the domain-wall propagation time because of a large pattern size (~500 nm). Therefore, the above estimates of the incubation delay is an upper bound and true delay must be even smaller. This value is much smaller than the STT incubation time that can be up to several tens of nanoseconds[30,31], indicating that the ultrafast SOT switching is achieved in our samples.

To better understand the short pulse induced SOT switching in the trilayer structure, we further carry out micromagnetic simulations in the presence of thermal fluctuation. Details of the parameters have been given in methods. Figure 4a shows the switching current density ($J_{sw}$) as a function of pulse width ($\tau_p$) with different spin polarization angles (η = 0.1 - 20°; (spin z)/(spin y) = tan(η)). Two observations are worth mentioning. First, $J_{sw}$ decreases with increasing spin polarization angles (η)[27], which influenced by the dynamics of domain nucleation, expansion, and the associated domain wall energy[35]. Given that a larger η indicates a larger ratio between the spin-z to spin-y polarization, the

$J_{sw}$ reduction with increasing η shows that the spin-$z$ spin current is more efficient to reduce $J_{sw}$ in comparison to the spin-$y$ component. The simulation results presented in Supplementary Fig. 11 provide additional insights into how the spin-$z$ polarization contributes to the reduction in $J_{sw}$. Second, except for the case with η = 0.1°, $J_{sw}$ increases with decreasing $\tau_p$, which is consistent with experimental results. This dependence of $J_{sw}$ on $\tau_p$ is caused by the fact that the lateral size of sample is large (i.e., R = 500 nm) and thus the switching is governed by domain nucleation and expansion. Figure 4b shows the ratio of $J_{sw}$ with $\tau_p$ = 0.5 ns to $J_{sw}$ with $\tau_p$ = 10 ns. This ratio is found to increase with η. Comparing to the ratio in experiment (about 3 to 4), this result suggests that the trilayer has a spin-$z$ polarization of about 6°.

In summary, we have demonstrated ultrafast and energy-efficient field-free SOT switching of PMA in FM trilayers. The switching time can be reduced to a sub-ns regime with a current density that is three to four times lower than the reported values of the field-assisted pulse switching. The spin-$z$ polarization in the FM trilayer structure is critical for the reduction of the switching current density, as has been confirmed by our micromagnetic simulations. The incubation time of the field-free SOT switching is estimated to be two to three orders smaller than the STT-driven switching which can be up to several tens of nanoseconds. Therefore, our work shows that the ultrafast SOT switching with the intrinsic field-free characteristic and a low switching current density is achieved in the magnetic trilayer structure, which is promising for reducing the energy and latency of the writing process in high-speed SOT-based logic and memory applications.

## Methods
### Sample preparation
Si/SiO$_2$ substrate/Ta (2 nm)/CoFeB (4 nm)/Ti ($t_{Ti}$ nm)/CoFeB (1 nm)/MgO (3.2 nm)/Ta (2 nm) structures with various Ti thicknesses ($t_{Ti}$ = 1 ~ 4 nm) were deposited by magnetron sputtering with a base pressure below $3.0 \times 10^{-8}$ Torr at room temperature. The metallic layers such as Ta, Ti and CoFeB were deposited using DC power with a working Ar pressure of 3 mTorr, while the insulating MgO layer was deposited using radio frequency (RF) power at 10 mTorr. To introduce in-plane magnetic anisotropy of the bottom CoFeB (4 nm) layer, an IP magnetic field of 15 mT was applied in the $x$ direction during the deposition. All samples were post-annealed at 150 °C for 40 mins to promote PMA of the top CoFeB (1 nm) layer. The film stacks then were fabricated into Hall bar devices with a FM pillar using electron-beam lithography and subsequent ion milling. The pillar was etched to the Ti layer and the layer thickness was further determined by atomic force microscope for current density calculation. The FM pillar radius (R) varies from 75 to 550 nm. Both the current and voltage channel widths are scaled proportionally according to the pillar size.

### DC-induced field-free SOT switching
A Keithley 6221 current source was utilized to apply pulsed DC currents (100 μs width) across the Hall channel and a Keithley 2182 A nanovoltmeter was used to record the resulting output Hall voltage. Before the measurement, an IP field of 100 mT was applied to saturate the magnetization of bottom CoFeB layer along the +$x$ or −$x$ direction to determine the field-free switching sequence.

### Short pulse induced field-free SOT switching
A short pulse was applied by a PSPL 10060 A pulse generator. Before the short pulse injection, we first saturated the magnetization of the bottom CoFeB layer to the +$x$ or -$x$ direction, and initialized the PMA of the up CoFeB layer to the 'up' or 'down' state. After each pulse injection, $R_{AHE}$ was probed to sense the perpendicular magnetization state with a low DC current to determine the switching probability using Keithley 6221 and 2182 A. Then the device was initialized again and we changed the pulse duration and repeated the pulse switching measurements. Each point was repeatedly measured 30 times and averaged out.

### Micromagnetic simulations
The damping-like torque (DLT) term, (d**M**/dt)$_{SOT}$ = ($\gamma c_J/M_s$) **M** × (**M** × **σ**), is adopted in the Landau-Lifshitz-Gilbert equation[27]. Here, $\gamma$ is the gyromagnetic ratio, **M** is the magnetization vector of the free layer (CoFeB/MgO PMA layers), **σ** is a unit vector along to the spin polarization, $M_s$ is the saturation magnetization, and $c_J = (\hbar\theta_D J/2eM_s t_z)$ is the DLT amplitude in the unit of magnetic field. $\hbar$ is the reduced Planck constant, **J** is the charge current density flowing along the $x$ axis, and $e$ is the electron charge. We assume that **σ** = (0, cos η, sin η) is a spin polarization direction and $\eta$ represents the spin-polarization angle. The following parameters are used for the free layer: the thickness ($t_z$) is 1 nm, the saturation magnetization $M_s$ is 1000 emu cm$^{-3}$, the exchange stiffness constant is $1.0 \times 10^{-6}$ erg cm$^{-1}$, the effective anisotropy field $H_{K,eff}$ is 400 mT, effective DLT efficiency $\theta_D$ is 0.3, the damping constant $\alpha$ is 0.005, and the unit cell size is 1 nm × 1000 nm × 1 nm. Spin polarization angle η is varied from 0.1 to 30 degree. An external magnetic field $H_x$ = 30 mT is applied only for η = 0.1°. The current pulse-width is varied from 0.5 to 10 ns with a rise/fall time of 0.2 ns. For the temperature effect, the Gaussian-distributed random fluctuation fields (mean = 0, standard deviation = $\sqrt{[2\alpha k_B T/(\gamma M_S V \Delta t)]}$, where $k_B$ is the Boltzmann constant, $\Delta t$ is the integration time step, $V$ is the volume of unit cell)[36] are added to the effective field of the Landau–Lifshitz–Gilbert equation. The switching probability as a function of current density is obtained with 200 different random seeds for Langevin thermal random field ($T$ = 300 K) and the switching current density $J_{sw}$ is determined to be the current density corresponding to the switching probability of 0.5.

## Data availability
The data that support the results of this study are available from the corresponding author upon reasonable request.

## Code availability
The codes that support this study can be available from the corresponding author upon reasonable request.

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

## Acknowledgements

The work is supported by SpOT-LITE programme (A*STAR grant, A18A6b0057) through RIE2020 funds, Singapore Ministry of Education (MOE) Tier 2 (R-263-000-E29-112), National Research Foundation (NRF) Singapore Investigatorship (NRFIO6-2020-0015) and Samsung Electronics Co., Ltd (IO221024-03172-01). B.-G.P. acknowledges support from the National Research Foundation of Korea (NRF-2022M3I7A2079267). K.-J.L. acknowledges support from the National Research Foundation of Korea (NRF- 2020R1A2C3013302).

## Author contributions

Q.Y. performed fabrications and measurements. D.H., J.K., and B.P. provided trilayer films. S.Z. helped short pulse measurements and gave suggestions on data analysis. F.W. performed atomic force microscope measurement. S.L., J.L., and K.L. carried out micromagnetic simulations. Q.Y., H.Y., K.L., and B.P., wrote the manuscript. The project was initiated by H.Y. All authors contributed to discussion of the results.

## Competing interests

The authors declare no competing interests.
