## [Peer Review File · Nature Communications]

Reviewers' Comments:

Reviewer #1:

Remarks to the Author:

This manuscript reports the field-free switching of ferromagnetic trilayers system in sub-ns timescale via the combination of spin-orbit torque and interface-induced perpendicular polarized spin current. The author fabricated similar multilayers as listed in ref 24 but adding one more Ta layer underneath to provide spin-Hall current. Sub-ns switching of top PMA layer was achieved by providing only fractions of current density that required from other similar field-free switching PMA devices. Anomalous Hall loop shift confirmed the existence of the perpendicular spin current in this heterostructure and the micromagnetic simulations are also consistent with the observed assistance effect on switching of PMA layer. In a nutshell, this manuscript provides an interesting and new idea to explore a fundamental physics question and potentially a method to reduce the switching current density of the PMA layer, which is of importance to realize practical spintronic applications such as MRAM and it will attract the audiences from the scientific community. However, after carefully reading the manuscript, the physics under this mixed interface/spin-Hall SOT switching devices is still not answered, and the interpretation of the data may not right. My concerns on the results need to be completely resolved before proceeding to the next stage.

1. In fig. 1 (b) and (c), the maximum AHE measured from out-of-plane magnetic field is much larger than the maximum value obtained from current sweeping loop. Is there any specific reason why this value is smaller in current sweeping loop? Is it means that the PMA layer is not fully saturated?

2. Please add x,y,z axes (both positive and negative) in fig. 1(a) schematic drawing for better readability.

3. The bottom in-plane magnetized FM layer will give an in-plane dipolar field on top PMA layer, which may also help the field-free switching. How much does this contribute to the overall efficiency?

4. The current sweeping loop in supplementary Fig. 1 is much noisy than M-H loop and some of them ($T_i = 2\text{nm}$ and 4nm) even has no change of sign in AHE signal at different polarity of the current. This makes me wonder if the PMA layer is switched or not and there is no clear trend when you are tailoring the different thickness of Ti. This weaken the claim that this perpendicular torque is generated from the interface of Ti/FM.

5. In fig. 1 (d) the anomalous Hall loop shift is not symmetric, not as clear as shown in ref. 24. Why is that? This cannot be explained by the spin-Hall current since it is also symmetric once you switch the magnetization direction. Notice that in the supplementary Fig. 2, the switching is discrete, will it affect your fitting? I am concerned about the sample quality, is it reproducible in similar devices? How about other samples with different Ti thickness?

6. Same asymmetry is observed in the pulse current switching measurements in Fig. 2 and also in other samples showed in the supplementary materials. Does it contribute to the variation of the sample geometry? How many samples you have measured?

7. From a more general perspective, the spin-orbit torque from spin-Hall effect is polarized along y-direction, while the magnetization is aligned along x-direction. Obviously, the spin-Hall current will exert a torque on the bottom in-plane layer. However, the patterned circle device comes with no shape anisotropy to help stabilize the magnetization. How strong is the anisotropy field of the bottom in-plane layer? Will it maintain stable at high current density? Is it the reason why you need to initialize the magnetization of the bottom layer? Will this break the symmetry of the system? Anyway to probe the possible change of the magnetization direction of this layer?

8. In micromagnetic study, the author claimed " J_{sw} ...", which is qualitatively consistent with the macrospin model", but later added "...this dependence of J_{sw} on τ_p is caused by... domain nucleation and expansion", giving two conflicted statements, please explain. It is more likely the device is partially switched due to the large lateral dimension. In this case, the energy barrier of

the switching is not the anisotropy energy anymore but becomes the domain wall energy. (see refs: APL Mater. 9, 091101(2021), APL. 100, 102401) This may give you a more accurate picture of the switching mechanism.

9. Since the author estimate the angle to be approximately 5 degree, is it agree well with ref. 24? From this reference, the interface perpendicular spin-orbit torque is sufficient to switch the PMA layer, how do you know if the spin-Hall current actually helped the switching process? Which one is the main contributor and how do you quantify it? One suggestion for the experiment: vary the thickness of the bottom in-plane FM layer and see how it weaken the spin-Hall current effect on the PMA switching. Besides, is it possible to make multilayers of (Ti/FM)_n to enhance this perpendicular spin current and make the angle larger than 5 degree? This may also give us a better understanding of your mixed IP and OOP spin current scenario.

Reviewer #2:

Remarks to the Author:

The article titled "Field-free spin-orbit torque switching in ferromagnetic trilayers at sub-ns timescales" by Qu Yang and colleagues reports on the achievement of sub-nanosecond field-free switching of CoFeB/Ti/CoFeB structures. These structures have one in-plane magnetized (IMA) CoFeB layer and one with out-of-plane anisotropy (PMA), separated by a thin Ti layer varying from 1 to 3nm. The aim is to extend these structures to three-terminal tunnel junction technology, where the reversal of the storage layer at zero external field, at nanosecond timescales and with low writing current, is currently a major challenge in the community.

In this work, the field-free switching is achieved by using a Z-spin polarization from the IMA CoFeB layer, while a Y-spin polarization ensures faster magnetization reversal due to instant torque and limited incubation delay. The authors demonstrate that in this simplified system, they can fulfill the above requirement with field-free switching at sub-nanosecond timescales and low current density. The authors also analyze the time dependence of the critical current and find that incubation delays are very low. Micromagnetic simulation supports the data. This demonstration holds exciting potential for the practical implementation of SOT-MRAM technology.

Overall, I find these results very interesting for the spintronics and microelectronics communities. The article is well-written and clear, and deserves to be published. However, I consider the work to be more incremental than novel, and I would not recommend it for publication in Nature Communications. Here are some reasons for my opinion.

Firstly, the proposed field-free approach has already been demonstrated in the literature in high-impact journals (e.g. ref. 24, which includes some authors of this manuscript). In this study, the authors extend the study of similar structures to nanosecond switching regimes, which helps to investigate the intrinsic switching current, incubation delay time, and device performance for memory applications in more detail.

On the other hand, the study focuses on micro-sized magnetic dots, and it is well-documented that reversal mechanisms and energies are very different in μm compared to sub-100 nm dots. Meanwhile, practical applications are clearly projected for sub-100 nm dimensions. Hence, despite the authors claim of very low switching current density, I think that some of the performance claims, and benchmarking may need to be lowered.

In Figure 2, the switching probabilities hardly converge to 100%, but they are not zero either. I raises some doubts about the switching reliability, and it would be helpful if the authors could comment on this. They should also provide P_{sw} vs. write current. Additionally, it would be interesting to document the switching when M is parallel to $+x$. Why is the study limited to M parallel to $-X$?

Regarding minor comments:

-In Table 1, the authors should specify how the current densities are estimated in each approach. They should also note that the writing current is dependent on the coercivity, which is much lower in these devices than in scaled SOT-MTJ reported in Table 1.

- Reference 35 given in table 1 is missing.

Reviewer #1 (Remarks to the Author):

This manuscript reports the field-free switching of ferromagnetic trilayers system in sub-ns timescale via the combination of spin-orbit torque and interface-induced perpendicular polarized spin current. The author fabricated similar multilayers as listed in ref 24 but adding one more Ta layer underneath to provide spin-Hall current. Sub-ns switching of top PMA layer was achieved by providing only fractions of current density that required from other similar field-free switching PMA devices. Anomalous Hall loop shift confirmed the existence of the perpendicular spin current in this heterostructure and the micromagnetic simulations are also consistent with the observed assistance effect on switching of PMA layer. In a nutshell, this manuscript provides an interesting and new idea to explore a fundamental physics question and potentially a method to reduce the switching current density of the PMA layer, which is of importance to realize practical spintronic applications such as MRAM and it will attract the audiences from the scientific community. However, after carefully reading the manuscript, the physics under this mixed interface/spin-Hall SOT switching devices is still not answered, and the interpretation of the data may not right. My concerns on the results need to be completely resolved before proceeding to the next stage.

Reply: We would like to thank the reviewer for his/her comments on the novelty and importance of our work. Taking into the consideration the reviewer's comments and suggestions we have improved our manuscript. We believe our paper now offers more sufficient and clear interpretation for the physics of the data. We hope that the manuscript is now suitable for publication.

1. In fig. 1 (b) and (c), the maximum AHE measured from out-of-plane magnetic field is much larger than the maximum value obtained from current sweeping loop. Is there any specific reason why this value is smaller in current sweeping loop? Is it means that the PMA layer is not fully saturated?

Reply: The field-free SOT switching portion of the sample in the original manuscript is about 75% compared to the magnitude of the AHE resistance. The partial switching may be attributed to the pinning of the domain walls at non-uniform edges of the nanopillar induced by physical etching. However, we find during the revision process that this ratio can be notably enhanced from 80% to 100% (shown in Figure R1) by employing a post-pillar etching step where the device is covered with a layer of Si₃N₄. We believe that this post-pillar etching process reduces the edge effects, resulting in the SOT-induce full magnetization switching without an external magnetic field.

For clearer understanding, we have added below sentences in Supplementary Note 1: "In most cases with FM trilayers, field-free magnetization switching of ~75% can be achieved. However, this switching ratio can be notably enhanced ranging from 80% to 100%, by employing a post-pillar etching step where the device is covered with a layer of Si₃N₄. We believe that this post-pillar etching process reduces the edge effects, resulting in the SOT-induce full magnetization switching without an external magnetic field."

Figure R1 | (a) Magnetic hysteresis loop and (b) Current induced field-free deterministic SOT switching in a R=200, 350, and 550 nm pillar devices. Nearly 100% switching can be achieved.

2. Please add x, y, z axes (both positive and negative) in fig. 1(a) schematic drawing for better readability.

Reply: Thanks for the comments. We have improved the schematic in Fig. 1a.

3. The bottom in-plane magnetized FM layer will give an in-plane dipolar field on top PMA layer, which may also help the field-free switching. How much does this contribute to the overall efficiency?

Reply: Thanks for pointing out an important question. The effect of the stray fields generated from the in-plane CoFeB on the field-free SOT switching in similar trilayer structures has been examined in a previous report [Adv. Mater. 34, 2203558 (2022)], where some of us are co-authors. It was concluded that the stray field is not strong enough to make field-free SOT switching by itself. In particular, the in-plane magnetic field induced by the bottom ferromagnetic layer acting on the top ferromagnetic layer was estimated to be ~ 0.1 mT. This value is much smaller than the effective in-plane magnetic field (2.7 mT) generated by the spin current in the magnetic trilayer structures [Adv. Mater. 34, 2203558 (2022)]. We have added related statement in the main text.

4. The current sweeping loop in supplementary Fig. 1 is much noisier than M-H loop and some of them ($t_{\text{Ti}} = 2$ nm and 4 nm) even has no change of sign in AHE signal at different polarity of the current. This makes me wonder if the PMA layer is switched or not and there is no clear trend when you are tailoring the different thickness of Ti. This weakens the claim that this perpendicular torque is generated from the interface of Ti/FM.

Reply: We feel sorry for presenting the noisy data in the original manuscript, where we more focused samples with $t_{\text{Ti}} = 3$ nm and paid less attention to the other samples. To respond the reviewer's comment, we have fabricated new devices with various Ti thicknesses and obtained consistent switching sequences with those in the original manuscript, which was updated in Supplementary Fig. 1. As shown in Figure R2 below, more distinct loops were clearly achieved.

In addition, we note that the interface-generated spin currents and associated switching behaviors in ferromagnet/Ti/CoFeB trilayers as a function of Ti thickness have been investigated in a previous report [Adv. Mater. Interfaces 9, 2201317 (2022)]. In such magnetic trilayers, in-plane (y-spin) and out-of-plane (z-spin) spin currents (or

resulting SOT) are generated from the spin-orbit filtering and precession of the interface-generated spin current, respectively. It was observed that the field-assisted SOT switching current strongly depends on t_{Ti} and even the SOT switching polarity changes as t_{Ti} increases. This indicates that the in-plane SOT is related to the relative current distribution between the bottom FM and Ti layers. In contrast, successful field-free SOT switching occurs only for small Ti thicknesses, up to 4 nm. This demonstrates that the out-of-plane SOT responsible for field-free switching is of interface origin. Furthermore, we have found that for a particular $t_{\text{Ti}} = 3$ nm, the field-free switching current is minimized. This suggests that the field-free switching in the magnetic trilayer is determined by a combination of in-plane and out-of-plane SOTs. We have provided above explanations in Supplementary Note 1.

Figure R2 | Field-free deterministic SOT switching as a function of current amplitude at different t_{Ti} , with \mathbf{M} saturated along the $+x$ and $-x$ direction. The devices have a pillar radius (R) of 500 nm and a channel width of 1.8 μm . J_c is 14.2 MA cm^{-2} for $t_{\text{Ti}} = 1$ nm (a), 15.7 MA cm^{-2} for $t_{\text{Ti}} = 2$ nm (b), and 32.5 MA cm^{-2} for $t_{\text{Ti}} = 4$ nm (c).

5. In fig. 1 (d) the anomalous Hall loop shift is not symmetric, not as clear as shown in ref. 24. Why is that? This cannot be explained by the spin-Hall current since it is also symmetric once you switch the magnetization direction. Notice that in the supplementary Fig. 2, the switching is discrete, will it affect your fitting? I am concerned about the sample quality, is it reproducible in similar devices? How about other samples with different Ti thickness?

Reply: The asymmetry seen in the anomalous Hall loop shift (previous Fig. 1d), in contrast to that in ref. 24, might be attributed to the utilization of smaller device dimension ($R=500$ nm) compared to the larger dimension ($R=4 \mu\text{m}$) in ref. 24. Due to the limitation of university nano-fabrication, it is unlikely to obtain a symmetric shaped sample after the lithography and etching processes. We have conducted AHE loop shift measurements using a larger sample ($R=2 \mu\text{m}$) and found a better symmetric behavior as shown in Figure R3 below.

Figure R3 | Current dependence of the anomalous Hall loop shift with M//+x and M//-x in a R=2 μm pillar device.

The discrete behavior shown in previous Supplementary Fig. 2 does not affect our fitting process as we did not extract any information from these switching loops. Using a new device, we also updated Supplementary Fig. 2 (Figure R4 below), which does not show any discrete behavior.

Figure R4 | Loop shift measurement under direct currents. **a**, AHE loops with $I_{dc} = \pm 0.1$ mA (M//-x). **b**, $I_{dc} = \pm 2.5$ mA (M//-x). **c**, $I_{dc} = \pm 2.5$ mA (M//+x). The sample is the same one in Fig. 1d.

As mentioned in Q4, we have tried different Ti thicknesses and the $t_{\text{Ti}} = 3$ nm sample shows the lowest switching current density. The field-free short pulse switching for different t_{Ti} is discussed in Supplementary Note 7. Regarding reproducibility, we have demonstrated repetitive field-free current switching in a single device (R=300 nm) without subsequent saturation of the bottom FM layer. This is evident in Figure R5a, which shows the sustained stable switching behavior even after more than one thousand cycles. We have also investigated switching behaviors across multiple devices and have observed consistent outcomes, as shown in Figures R4b and R4c below. Notably, we found that devices made from films with identical layer thicknesses maintain their reproducibility, even using different film batches (also discussed in Q6). We have incorporated relevant statements in Supplementary Note 4.

Figure R5 | **a**, Repetitive field-free switching in a single device ($R=300$ nm, $W=800$ nm) without subsequent saturation of the bottom FM layer. **b**, AHE loop. **c**, Field-free switching across multiple devices.

6. Same asymmetry is observed in the pulse current switching measurements in Fig. 2 and also in other samples showed in the supplementary materials. Does it contribute to the variation of the sample geometry? How many samples you have measured?

Reply: We thank for the comments. We conducted measurements on samples of varying dimensions and from different production batches with $t_{Ti} = 3$ nm. The results turn out that asymmetry is a common phenomenon in short pulse switching. Sample geometry can be a possible cause of this asymmetry as the etching process can unavoidably induce device asymmetry and the asymmetric domain nucleation takes place at the edge of the sample.

Figure R6 (updated supplementary Fig. 5) show the data for $t_{Ti} = 3$ nm obtained from another pillar device (device 2, $R=500$ nm). Similar to the findings for $M//x$ shown in Fig. 2, an asymmetry has also been observed when applying both positive and negative pulses. To establish one dataset in one device, we repeated measurements 5 times. Devices made from films with identical layer thicknesses maintain their reproducibility, as evidenced by the DC switching (Supplementary Note 4), short pulse switching (Supplementary Note 5) with various dimensions ($R=75\sim 550$ nm, Supplementary Note 6) and with different t_{Ti} (Supplementary Note 7).

Figure R6 | **a,b**, Short pulse field-free SOT switching probability as a function of (a) positive and (b) negative pulse duration at $t_{Ti} = 3$ nm from another pillar device ($R=500$ nm) when $M//x$. **c**, $|J_c|$ for $P = 0.9$ in two devices.

7. From a more general perspective, the spin-orbit torque from spin-Hall effect is polarized along y-direction, while the magnetization is aligned along x-direction. Obviously, the spin-Hall current will exert a torque on the bottom in-plane layer. However, the patterned circle device comes with no shape anisotropy to help stabilize the magnetization. How strong is the anisotropy field of the bottom in-plane layer? Will it maintain stable at high current density? Is it the reason why you need to initialize the magnetization of the bottom layer? Will this break the symmetry of the system? Anyway to probe the possible change of the magnetization direction of this layer?

Reply: Thanks for this important comment. The pillar is etched to the Ti layer only (as shown in Fig. 1a and b), and the bottom FM layer is fully covered across the Hall bar channel. To introduce in-plane (IP) magnetic anisotropy of the bottom CoFeB (4 nm) layer, an IP magnetic field of 15 mT was applied in the x-direction during the deposition (Methods section). Before fabrication, we use VSM to probe the anisotropy field of the bottom in-plane layer, as shown in Figure R7a.

We need to initialize the magnetization of the bottom FM layer (\mathbf{M}) because it determines the direction of spin polarization ($\mathbf{M} \times \mathbf{y}$) when charge flows along \mathbf{x} . By initializing the bottom magnetic layer in different directions, we can control the switching polarity of the top magnetic layer (Fig. 1c). The spin currents generated by the bottom layer or its interfaces will exert a torque on the magnetization of the top FM layer, attempting to align it with the spin current direction. The specific initialization direction of the bottom FM layer will determine the preferred alignment direction for the top layer during the switching process. The initialization of \mathbf{M} will break the symmetry of the system. As mentioned above, the initialization of the bottom FM layer in a specific direction sets the reference state for the magnetization configuration, which influences the subsequent spin current and torque direction. We have given related statements in the main text.

We conducted an endurance test for \mathbf{M} stability by repeatedly performing field-free DC switching more than a thousand cycles without subsequent saturation of the bottom FM layer, as shown in Figure R7b (also Figure R5a in Q5, and updated Supplementary Fig. 4). As discussed earlier and also indicated in Fig. 1c, the switching polarity of the top PMA is determined by \mathbf{M} . For instance, in the case of $\mathbf{M} // +x$, the polarity of field-free switching should be clockwise, and anti-clockwise if \mathbf{M} switches to $-x$. The results demonstrate persistent switching polarity even after 1200 cycles without subsequent saturation of the bottom FM layer, suggesting the robustness of \mathbf{M} during pulse switching.

Nevertheless, we agree with the reviewer's comment that the magnetization of the bottom FM will ultimately deviate from the x-direction after many switching cycles due to the torque generated from the top CoFeB and Ta layers, thereby deteriorating the field-free switching characteristics. This can be overcome by introducing an antiferromagnetic layer exchange-coupled with the bottom ferromagnet to keep it as a single domain as demonstrated in the previous report [Adv. Mater. 34, 2203558 (2022)].

Figure R7 (a) Magnetic hysteresis loops of $t_{\text{Ti}} = 3$ nm trilayer sample along different directions and the IP anisotropy field of bottom FM layer is ~ 17 Oe. (b) Repetitive field-free current switching in a single device ($R=300$ nm, $W=800$ nm) to test M stability.

8. In micromagnetic study, the author claimed “switching current density is qualitatively consistent with the macrospin model”, but later added “this dependence of switching current density on pulse width is caused by domain nucleation and expansion”, giving two conflicted statements, please explain. It is more likely the device is partially switched due to the large lateral dimension (1000 nm). In this case, the energy barrier of the switching is not the anisotropy energy anymore but becomes the domain wall energy. (see refs: APL Mater. 9, 091101(2021), APL. 100, 102401) This may give you a more accurate picture of the switching mechanism.

Reply: Thanks for the comments. We acknowledge that the energy barrier for switching is primarily influenced by the domain wall energy rather than the anisotropy energy. We have made the necessary revisions to enhance clarity in this regard and cited the relevant references the referee mentioned.

9. Since the author estimate the angle to be approximately 5 degree, is it agree well with ref. 24? From this reference, the interface perpendicular spin-orbit torque is sufficient to switch the PMA layer, how do you know if the spin-Hall current actually helped the switching process? Which one is the main contributor and how do you quantify it? One suggestion for the experiment: vary the thickness of the bottom in-plane FM layer and see how it weaken the spin-Hall current effect on the PMA switching. Besides, is it possible to make multilayers of (Ti/FM) $_n$ to enhance this perpendicular spin current and make the angle larger than 5 degree? This may also give us a better understanding of your mixed IP and OOP spin current scenario.

Reply: Ref. 24 discussed y- and z-spins in Fig. 3a, however they didn’t calculate the tilting angle. Given the ratio of z-spin/y-spin as 0.1~0.2, we can compute a spin-z polarization angle ranging from 5.7 to 11 degrees. This calculation aligns with our estimated angle of 6 degrees.

Concerning the spin-Hall current originating from the bottom Ta layer, it is important to note that this in-plane SOT is primarily influenced by an applied magnetic field [Nat. Electron. 5, 217 (2022)]. However, in the absence of an external field, in-plane SOT cannot switch the magnetization direction alone without out-of-plane SOT [Adv. Mater. Interfaces, 9, 2201317 (2022)]. Our micromagnetic simulations also have

confirmed that in-plane SOT alone (Fig. 4a, $\eta=0.1^\circ$) is insufficient for achieving PMA switching. Therefore, we conclude that the out-of-plane spins are the main contributor of field-free switching. We have given related statements in Supplementary Note 1.

The influence of the bottom CoFeB thickness on field-free SOT switching was explored in a previous report [Adv. Mater. Interfaces, 9, 2201317 (2022)]. The results show that there is no field-free SOT switching in the CoFeB/Ti/CoFeB sample with a thicker CoFeB (> 6 nm). Moreover, no field-free SOT switching occurs in samples with a large t_{Ti} (> 4 nm). These results suggest that out-of-plane SOT mainly originates from the CoFeB/Ti interface. Besides, it is reported that a straightforward insertion of a 1-nm-thick NiFe or CoFeB layer between the in-plane FM and Ti layers (with a total FM thickness of 4 nm) leads to a complete reversal of the field-free SOT switching behavior. [Nat. Mater. 17, 509-513 (2018)]. The results indicate that the sign of the SOT is determined by the thinner (1 nm) inserted layer rather than the thicker (3 nm) bottom FM layer, and it should be independent of the bulk SHE effect in the bottommost Ta layer. We have added related statements in Supplementary Note 1.

Regarding the enhancement of spin-z polarization angle (η), we have performed micromagnetic simulations on Mumax3 [AIP Adv. 4, 107133 (2014)] to evaluate the effect of η on SOT switching. As shown in Figure R8, for pure y -spin (blue star, $\eta=0^\circ$), the symmetry-breaking magnetic field along the x -axis is essential for switching, with an estimated switching current (J_{sw}) of 46.5 MA cm^{-2} . In the case of z -spin (red dots, field-free), introducing a 5° spin- z polarization leads to a 30% reduction in J_{sw} , while a 10° spin- z polarization results in a 42.5% decrease in J_{sw} . We have added the above statements in the main text and Supplementary Note 10.

Figure R8 | Macrospin simulations of DC-induced SOT switching for different spin- z polarization angles. J_{sw} for different η at zero field (red dots). The blue star shows J_{sw} for $\eta = 0^\circ$ (pure spin y) with the external magnetic field $H_x = 200 \text{ Oe}$.

Reviewer #2 (Remarks to the Author):

The article titled "Field-free spin-orbit torque switching in ferromagnetic trilayers at sub-ns timescales" by Qu Yang and colleagues reports on the achievement of sub-nanosecond field-free switching of CoFeB/Ti/CoFeB structures. These structures have one in-plane magnetized (IMA) CoFeB layer and one with out-of-plane anisotropy (PMA), separated by a thin Ti layer varying from 1 to 3nm. The aim is to extend these structures to three-terminal tunnel junction technology, where the reversal of the storage layer at zero external field, at nanosecond timescales and with low writing current, is currently a major challenge in the community.

In this work, the field-free switching is achieved by using a Z-spin polarization from the IMA CoFeB layer, while a Y-spin polarization ensures faster magnetization reversal due to instant torque and limited incubation delay. The authors demonstrate that in this simplified system, they can fulfill the above requirement with field-free switching at sub-nanosecond timescales and low current density. The authors also analyze the time dependence of the critical current and find that incubation delays are very low. Micromagnetic simulation supports the data. This demonstration holds exciting potential for the practical implementation of SOT-MRAM technology.

Overall, I find these results very interesting for the spintronics and microelectronics communities. The article is well-written and clear, and deserves to be published. However, I consider the work to be more incremental than novel, and I would not recommend it for publication in Nature Communications. Here are some reasons for my opinion.

Reply: We sincerely appreciate the reviewer's positive comments regarding the clarity and potential impact of our findings on the spintronics and microelectronics communities. The reviewer's valuable questions regarding the novelty, device dimension, switching reliability, and the need for additional experiments have greatly contributed to the improvement of our work. We have carefully addressed each question in our responses and made necessary revisions to provide a clearer understanding. We kindly request to reconsider for publication.

1. Firstly, the proposed field-free approach has already been demonstrated in the literature in high-impact journals (e.g. ref. 24, which includes some authors of this manuscript). In this study, the authors extend the study of similar structures to nanosecond switching regimes, which helps to investigate the intrinsic switching current, incubation delay time, and device performance for memory applications in more detail.

Reply: Dampinglike SOT in our samples includes both $\mathbf{m} \times (\mathbf{m} \times \mathbf{y})$, originating from spin-y spin currents, and $\mathbf{m} \times (\mathbf{m} \times \mathbf{z})$, originating from spin-z spin currents. As $\mathbf{m} \times (\mathbf{m} \times \mathbf{z})$ torque has the same form as the conventional spin-transfer torque in perpendicular MRAMs, it causes many precessions during the switching, which is shown in Fig. 1 of Sci. Rep. 10, 1772 (2020). It is known that such a precession motion increases the switching current at short pulses and this increase depends on the relative ratio of spin-z spin current to spin-y spin current [see Fig. 3 of Sci. Rep. 10, 1772 (2020)]. Therefore, to find out if the field-free SOT switching is efficient in terms of power consumption,

it is of crucial importance to experimentally measure the switching current at short pulses. This was carried out in this study.

In our study, we achieved sub-ns field-free switching of PMA with significantly lower J_c compared to reported field-assisted pulse switching at similar timescales. During this rebuttal, we further performed short pulse switching in devices with varying radius (R), ranging from 75 to 550 nm. The minimum pulse width required to achieve $P_{sw} = 0.9$ for the field-free SOT switching can be as short as $\tau_p = 0.3$ ns for R=500 nm. With a reduced radius of R=300 nm, τ_p can be further decreased to just 0.14 ns. The estimated incubation time of our switching process has been further reduced to a range of 0.0144 to 0.226 ns. This is notably shorter than the incubation time associated with STT switching, which can extend up to several tens of nanoseconds. Therefore, our work on ultrafast SOT switching offers substantial contributions to the field. By delving into ultrafast dynamics, our study enhances the understanding of fundamental physics and technological implications, complementing the foundation laid by previous DC field-free SOT demonstrations [ref. 24, Nat. Mater. 17, 509-513 (2018)].

2. On the other hand, the study focuses on micro-sized magnetic dots, and it is well-documented that reversal mechanisms and energies are very different in μm compared to sub-100 nm dots. Meanwhile, practical applications are clearly projected for sub-100 nm dimensions. Hence, despite the authors claim of very low switching current density, I think that some of the performance claims, and benchmarking may need to be lowered.

Reply: We totally agree with the reviewer. Recognizing the importance of investigating the influence of device size on our findings and its practical implications, we conducted additional experiments using smaller devices. As shown in Figure R9, short pulse switching was performed in devices with varying R, ranging from 75 to 550 nm. In smaller devices, the influence of edges and interfaces becomes more significant, resulting in an increase in J_c and writing energy, as summarized in Fig. R8e,f. We have incorporated Fig. R9 in Supplementary Note 6.

Figure R9 | Short pulse field-free SOT switching probability with (a) $R=550$ nm, (b) $R=300$ nm, and (c) $R=150$ nm with $M//+x$. d, J_c for $P = 0.9$ as a function of τ_p with varying R . e, J_c as a function of R with DC, $\tau_p=1$ ns, 3 ns. f, Writing energy as a function of R .

3. In Figure 2, the switching probabilities hardly converge to 100%, but they are not zero either. I raises some doubts about the switching reliability, and it would be helpful if the authors could comment on this.

Reply: We would like to thank the reviewer for pointing this out. In our previous analysis, the normalization approach using maximum and minimum R_{AHE} values posed challenges in achieving 100% or 0%. To overcome this, we propose a more effective normalization method utilizing the average R_{AHE} values of both the 'down' and 'up' states. These improvements enhance the accuracy and reliability of our measurements. We have updated Fig. 2 (also Figure R10 below).

Figure R10 | Field-free short pulse SOT switching in CoFeB/Ti (3 nm)/CoFeB trilayer with $R = 500$ nm. A single positive pulse (a) or negative pulse (b) is applied as a function of τ_p at different pulse current densities. M is saturated along the $-x$ direction.

Regarding the switching reliability, we fabricated multiple devices within same chip, and observed that similar switching behaviors can be achieved across these devices. Moreover, we have found that the reproducibility of devices made from films with the same layer thickness remains stable, even when the films originate from different batches (Supplementary Note 4).

Figure R11 shows supplementary data for the $t_{Ti} = 3$ nm sample obtained from another pillar device (device 2, $R=500$ nm), which is similar to the findings for $M//x$ shown in Fig. 2. To establish one dataset in one device, we repeated these measurements 5 times. Moreover, field-free short pulse switching can be repeated in different devices (Supplementary Note 5), across various dimensions ($R=75\sim 550$ nm, Supplementary Note 6), and with different t_{Ti} (Supplementary Note 7).

Figure R11 | Short pulse field-free SOT switching probability as a function of (a) positive and (b) negative pulse duration at $t_{Ti} = 3$ nm from another pillar device ($R=500$ nm) when $M//x$. (c) Comparing $|J_c|$ for $P = 0.9$ as a function of τ_p in two devices.

4. They should also provide P_{sw} vs. write current.

Reply: Thanks for the comment. Figure R12 shows the switching probabilities as a function of write current at different pulse widths for $t_{Ti} = 3$ nm. In DC switching, the current density tends to be relatively low, whereas short pulse switching demands much higher current densities as the pulse width decreases to the sub-ns region. We have added related statements and figure in Supplementary Note 8.

Figure R12 | Short pulse and DC induced field-free SOT switching probabilities as a function of current density for $t_{Ti} = 3$ nm. The device with $R=500$ nm is the same one in Fig. 1c and Fig. 2a.

5. Additionally, it would be interesting to document the switching when M is parallel to $+x$. Why is the study limited to M parallel to $-x$?

Reply: We acknowledge the importance of studying both orientations comprehensively to gain a complete understanding of the system behavior. Now we have updated the manuscript, and the results of $M//+x$ has also been added in Supplementary Fig. 6 ($R=75\sim 550$ nm) and Supplementary Fig. 7 (different t_{Ti} , also Figure R13 below) and related statements also have been added in main text and Supplementary Note. By initializing M in different directions, we can control the switching polarity of the top magnetic layer (Fig. 1c).

Figure R13 | Short pulse field-free SOT switching probability as a function of pulse duration when $M//+x$. **a,b**, Pulse switching probability (P_{sw}) for $t_{Ti} = 1$ nm with positive pulses (**a**) and negative pulses (**b**). **c,d**, P_{sw} for $t_{Ti} = 3$ nm with positive pulses (**c**) and negative pulses (**d**).

6. In Table 1, the authors should specify how the current densities are estimated in each approach. They should also note that the writing current is dependent on the coercivity, which is much lower in these devices than in scaled SOT-MTJ reported in Table 1.

Reply: Thanks for the comments. The short pulse switching current densities can be found or estimated from the cited papers and how the current densities are estimated is shown below:

-Ref. 15: J_c can find in Fig.4.

-Ref. 27: J_c for 10 ns SOT switching has been given in the paper as well as the critical energy ($E_c = V_c^2 \tau_p / R$) at 0.27 ns and 10 ns can be find in Fig. 2f. Then we can calculate J_c at 0.27 ns.

-Ref. 35 (new Ref. 37): The intrinsic SOT current density (J_{c0}) is 140 MA cm⁻² at $V_{c0} = 0.39$ V (state in paper). Based on Fig. 4, we can calculate the value of J_c at 0.6 ns with $V_c = 0.769$ V.

-Ref. 28: J_c can be find in Fig. 3a.

Upon reviewing the citations again and using our new data in this revision, we have made slight updates in Table 1. In terms of short pulse switching current density, it was hard to find a clear trend in relation to coercivity. Indeed, the switching current should be influenced by the pulse width, device dimensions, material properties, contact design, impedance matching and the high-frequency experimental setups. Thus, it is not easy to compare apples to apples. Rather, we added one more column to provide B_{an} information, as the perpendicular anisotropy field (B_{an}) competes with switching current [Phys. Rev. Lett. 109.9 (2012): 096602; Appl. Phys. Lett. 118, 092406 (2021)].

Device Structure (nm)	Pulse width (ns)	Anisotropy field B_{an} (mT)	J_c (MA cm ⁻²)	Magnetic field (mT)	Write energy per area (mJ cm ⁻²)	Reference
Pt(3)/Co(0.6)	0.3	1000	387.5	91	11.2	15
W(3.7)/CoFeB(0.9)	0.27	270	312	23	8	27
W/CoFeB(0.9)	0.6	--	276	40*	19.1	37
Ta(10)/CoFeB(1)	0.4	--	340	100	2971	28
CoFeB(4)/Ti(3)/CoFeB(1)	0.14	400	73.2	0	1.51	This work

7. Reference 35 given in table 1 is missing.

Reply: Thanks for the comment. We have included Reference 35 (new ref. 37) in the reference list.

Reviewers' Comments:

Reviewer #1:

Remarks to the Author:

The author addressed all my comments and provides enough additional experiments for some important points, especially the z-component spin current detection and size-dependent study. I think the paper now is more appropriate for the publication.

Reviewer #2:

Remarks to the Author:

In my opinion, the authors have properly addressed the question raised by both reviewers, and I have no specific comments, except one, see below. They have also conducted consequent additional set of measurement, which are improving the manuscript content and information, and I even personally think that some of the results on the scaling deserve to be added in the main manuscript.

Remark: When looking at figure 3.b, the fitting of J_c vs. $1/t_p$ is not working very well and the conclusion taken from this remain therefore blur. This suggest that there may a need of revised interpretation due to interplay of Y and Z spin torques, or some experimental correction to be conducted (although it seems that pulse amplitude vs. t_p was appropriately calibrated). These experimental deviations should not limit the publication, but call clarification.

In conclusion, the study is rich and well documented, and the work of great interest for spintronic and MRAM communities, and deserves publication as is. However, I still find the work incremental with respect to already existing literature and hence believe that publication in Nature Communication is not exactly fitting with the journal scope.

Reviewer #2 (Remarks to the Author):

In my opinion, the authors have properly addressed the question raised by both reviewers, and I have no specific comments, except one, see below. They have also conducted consequent additional set of measurement, which are improving the manuscript content and information, and I even personally think that some of the results on the scaling deserve to be added in the main manuscript.

Remark: When looking at figure 3.b, the fitting of J_c vs. $1/t_p$ is not working very well and the conclusion taken from this remain therefore blur. This suggest that there may a need of revised interpretation due to interplay of Y and Z spin torques, or some experimental correction to be conducted (although it seems that pulse amplitude vs. t_p was appropriately calibrated). These experimental deviations should not limit the publication, but call clarification.

Reply: Thanks for the comment. Analytic models for finite temperature spin-torque dynamic have primarily focused on uniaxial single domain nanomagnets [Appl. Phys. Lett. 97, 262502 2010; Appl. Phys. Lett. 105, 212402 (2014)]. In our specific case, the device is more likely exhibiting multi-domain wall motion rather than reversing as a single magnetic domain. This characteristic may contribute to the observed deviations from the linearity of the experimental results. We have made the necessary statements in the main text to clarify this regard.

In conclusion, the study is rich and well documented, and the work of great interest for spintronic and MRAM communities, and deserves publication as is. However, I still find the work incremental with respect to already existing literature and hence believe that publication in Nature Communication is not exactly fitting with the journal scope.